

# FAME (v1.0): a simple module to simulate the effect of planktonic foraminifer species-specific habitat on their oxygen isotopic content

Didier M. Roche[1,2], Claire Waelbroeck[1], Brett Metcalfe[1,2], and Thibaut Caley[3]

[1]Laboratoire des Sciences du Climat et de l'Environnement, LSCE/IPSL, CEA-CNRS-UVSQ, Université Paris-Saclay, F-91191 Gif-sur-Yvette, France
[2]Vrije Universiteit Amsterdam, Faculty of Earth and Life Sciences, Cluster Earth and Climate, de Boelelaan 1085, 1081HV Amsterdam, The Netherlands
[3]EPOC, UMR 5805, CNRS, University Bordeaux, Pessac, France

*Correspondence to:* Roche, D. M. (didier.roche@lsce.ipsl.fr)

**Abstract.** The oxygen-18 signal recorded in fossil planktonic foraminifer shells has been used for over 50 years in many geoscience applications. However, different planktonic foraminifer species generally yield distinct oxygen-18 signals, as a consequence of their specific living habitats in the water column and along the year. This complexity is usually not taken into account in data – model integration studies. To overcome this shortcoming, we developed the FAME (Foraminifers As

Modeled Entities) module. The module predicts the presence or absence of commonly used planktonic foraminifers, and their oxygen-18 values. It is only forced by hydrographic data and uses a very limited number of parameters, almost all derived from culture experiments. FAME performance is evaluated using MARGO Late Holocene planktonic foraminifer calcite oxygen-18 and abundances data sets. The application of FAME to a simple cooling scenario demonstrates its utility to predict changes in planktonic foraminifer oxygen-18 in response to changing climatic conditions.

**1 Introduction**

Since the early work of Emiliani (1955), oxygen-18 isotopic abundance in calcite fossil foraminifer tests recovered from oceanic sediments has been widely used to reconstruct the past variations in oxygen-18 content of seawater as well as its temperature, the two main variables that affect the content of oxygen-18 in calcite. The recognition that different species of foraminifers from the same sediment-core yielded different oxygen-18 was made early on (e.g. Duplessy et al., 1970; Lidz

et al., 1968; Berger, 1969; Fairbanks and Wiebe, 1980; Deuser, 1987), though it was (Emiliani, 1954, 's) attempt to relate depth habitat of foraminifers to the density of seawater that led to the revelation that the oxygen-18 value recorded by fossil foraminifers likely favoured the average depth habitat of individual species. Through in situ water column sampling via opening-closing plankton nets (Jones, 1967) corroborated the depth habitats of Emiliani (1954). However, increased plankton sampling (Bé and Tolderlund, 1971) and the advent of the sediment trap have shown that different species have different living

habitats in the water column and along the year and that in some cases the foraminiferal oxygen-18 content presents an offset with respect to equilibrium calcite oxygen-18 (Mix, 1987; Bijma and Hemleben, 1994; Ortiz et al., 1995; Kohfeld et al., 1996; Bauch et al., 1997; Schiebel et al., 2002; Simstich et al., 2003; Mortyn and Charles, 2003; Rebotim et al., 2017; Jonkers and





Kucera, 2015). This complexity is usually not accounted for in paleoceanographic studies. Instead, the approximation is often made that each planktonic foraminifer species has an apparent living depth – defined as the water depth where equilibrium calcite formation would approximate their measured calcite oxygen-18 value in the water column – that can vary by hundreds of meters from one region to another. To correctly interpret the wealth of information coming from the calcite oxygen-18

record, especially when multiple species are measured at the same geographical location, there is a need to take into account the impact of depth habitat and growth season on each species calcite oxygen-18. Minor contributors to the resultant calcite oxygen-18 signal, such as carbonate ion concentration (Spero et al., 1997) and symbionts (e.g. Ezard et al., 2015; Spero, 1998) may modulate the absolute values making species specific comparisons problematic, however their overall contribution may also covary and/or auto-correlate with temperature and latitudinal gradients, therefore this paper focuses on the major compo-

nents only.

     In recent years, the development of water isotope enabled ocean models has allowed the simulation of the two variables at the root of the calcite oxygen-18 record: seawater temperature and oxygen-18. It is thus tempting to make one additional step and attempt to compute a calcite oxygen-18 content that can readily be compared with the foraminiferal record. So far, oxygen-18

model-data studies have generally compared planktonic oxygen-18 to equilibrium calcite oxygen-18 values. The equilibrium calcite oxygen-18 in that case is computed from annual averaged seawater temperatures and oxygen-18 taken either at surface or averaged over the upper 50 to 100 meters of the water column (e.g. Caley et al., 2014; Werner et al., 2016). To go one step further, it is necessary to account for species-specific habitat when computing calcite oxygen-18. The FAME (Foraminifers As Modeled Entities) approach is underpinned by two arguably simple principles: 1) that the weighting due to species-habitat is

reflected in the calcite oxygen-18 record and 2) that the model derived should be kept simple to allow its offline application to the output of climatic models without the need of re-running the entire climate model simulations.

     After having developed the FAME methodology, we found out that the idea was already present in a theoritical framework in Mix (1987) and in one following study (Mulitza et al., 1997), in the latter referred to as *Mix's model*. The most notable

difference between the early study of Mix (1987) and the present one is the actual definition of the weighting functions. Mix (1987) assumed them being simple Gaussians whereas we build ours on the laboratory culture-based equations of planktonic foraminifer growth rates as a function of temperature given in Lombard et al. (2009).

     Since the early work of Mix (1987), other methods were developed to approach the species-specific complexity of planktonic

foraminifers. Schmidt (1999) developed a simple module to compute planktonic foraminifer oxygen-18 in an oxygen-18 enabled global ocean model. However, in his approach, water depths at which planktonic foraminifers calcify and their seasonal growth patterns are fixed for each species. Therefore, such a module can not properly account for the impact of climatic changes on foraminifer living conditions. Fraile et al. (2008) and Lombard et al. (2011) developed models predicting the abundance of common planktonic foraminifer species in response to hydrographic data and food concentration. Both these models predict

the relative abundances of the different simulated foraminifer species, an information which is not needed to assess individual





species oxygen-18 but entails a large number of empirical parameters, i.e. 21 and 15 parameters per planktonic foraminifer species in Fraile et al. (2008) and Lombard et al. (2011) respectively. Moreover, Fraile et al. (2008) derive the sensitivity of each species with respect to temperature from sediment-trap data, so that their model can only account for changes in seasonality, and not in depth habitat. In contrast, the FORAMCLIM model (Lombard et al., 2011) predicts both season and water depth

of each species potential maximum abundance. In fact, FAME can be viewed as a simplified version of FORAMCLIM (only retaining FORAMCLIM's computation of growth rates as a function of temperature), expanded by a mechanistic calculation of species-dependent calcite oxygen-18. FAME is only forced by hydrographic data, and only uses 6 parameters per planktonic foraminifer species that are all derived from culture experiments, plus one parameter accounting for the effect of the accretion of a calcite crust by N. pachyderma. Taken together, these characteristics make FAME a uniquely simple and robust model

designed to predict changes in the oxygen-18 of commonly used planktonic foraminifers in response to changing climatic conditions.

## 2 Methodology

The calcite oxygen-18/oxygen-16 ratio ($\delta^{18}O_c$, in per mil versus V-PDB in what follows) of planktonic foraminifers is intrisi-

cally a 4-dimensional signal, acquired at a specific season (time dimension), over a specific depth range and area in the ocean (space dimensions). The mean $\delta^{18}O_c$ signal measured on a sample composed of a number of individual foraminifer shells of one species is thus the integration of many different single $\delta^{18}O_c$ paths in this 4-dimensional space. If we suppose that the sampled population is representative of the living conditions of the species, it is thus likely that there is an over-sampling of the areas and time representing favourable growth conditions and an under-sampling of area and time with unfavourable growth

conditions. Hence, a reasonable way to predict the mean $\delta^{18}O_c$ of a foraminifer sample constituted of a number of individuals is to weight the oceanic conditions by the growth rate of each individual. The predicted $\delta^{18}O_c$ is then a weighted sum of these conditions in space and time.

### 2.1 Basic equations

To define the effect of the habitats of the different foraminifer species, we first consider a subset of the growth functions derived by Lombard et al. (2009) from culture experiments (Figure 1). For each foraminifer species $k$ considered, the growth function is written as:

$$\mu\left(T,k\right) = \frac{\mu\left(T_1,k\right) \cdot exp\left(\frac{T_A}{T_1} - \frac{T_A}{T}\right)}{1 + exp\left(\frac{T_{AL}(k)}{T} - \frac{T_{AL}(k)}{T_L(k)}\right) + exp\left(\frac{T_{AH}(k)}{T_H(k)} - \frac{T_{AH}(k)}{T}\right)} \tag{1}$$





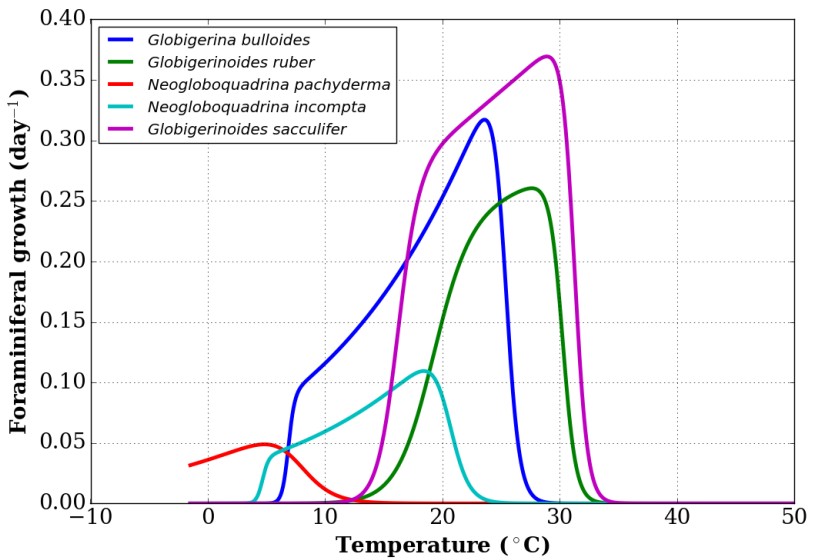

**Figure 1.** Growth functions corresponding to equation (1) over the full temperature range.

where $\mu(T,k)$ is the growth rate at temperature $T$ for the species $k$, $\mu(T1,k)$ is the growth rate for a chosen reference temperature T1 (20°C or 293K), $T_A$ is the Arrhenius temperature, $T_H(k)$ and $T_L(k)$ define the upper and lower boundaries of the growth tolerance range for the species $k$, $T_{AH}(k)$ and $T_{AL}(k)$ the Arrhenius temperatures for the decrease in growth rate respectively above and below these boundaries for species $k$ (Lombard et al., 2009). In the present study, we use the nominal

values of equation (1) parameters given in Lombard et al. (2009) with one exception: we use TL = 280K instead of 281.1K for *G. bulloides* in order to improve the representation of the seasonal cycle. When compared with sediment trap data from the subpolar North Atlantic (Jonkers et al., 2013), the use of nominal values does indeed lead to no growth outside of July, August and September, whereas the data show fluxes larger than 5 specimens per m$^2$ per day from the end of June to the middle of November, on average over the four years of observations.

We compute the $\mu(T,k)$ coefficient for all values of $T(x,y,z,t)$ in the world ocean, $T$ being a 4-D variable of space and time. This, in turn gives us the growth rate of the different foraminifer species considered in a 4-dimensional space as:

$$\mu(T,k) = \mu(T(x,y,z,t),k) \tag{2}$$

To avoid numerical issues in the code, we limit the value of $\mu(T,k)$ on the low end as follow:

$\mu'(T(x,y,z,t),k) = \mu(T,k)$        if  $\mu(T,k) \geq 0.1 \cdot \max_{T} \mu(T,k)$

$= 0$                otherwise                                  (3)

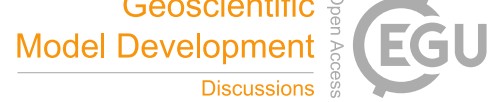



Given a 4-dimensional input field for oceanic temperatures and $\delta^{18}O$ of seawater, the equilibrium inorganic calcite $\delta^{18}O$ value can be computed from the temperature equation of Kim and O'Neil (1997). Here we use the quadratic approximation of that equation given in (Bemis et al., 1998):

$$T = T_0 - b \cdot \left(\delta^{18}O_c - \delta^{18}O_{sw}\right) + a \cdot \left(\delta^{18}O_c - \delta^{18}O_{sw}\right)^2 \tag{4}$$

where $T_0 = 16.1°C$, $b = -4.64$ and $a = 0.09$, $\delta^{18}O_{sw}$ is the seawater $\delta^{18}O$. Since the seawater temperature (°C) and $\delta^{18}O_{sw}$ (per mil) are inputs, we can solve this equation to determine the value of $\delta^{18}O_c$.

With the discriminant of the second degree equation being:

$$\Delta = b^2 - 4a \cdot (T_0 - T_{sw}) \tag{5}$$

it becomes:

$$\delta^{18}O_{c,eq} = \frac{-b - \sqrt{\Delta}}{2a} + \delta^{18}O_sw - 0.27 \tag{6}$$

where the constant, $0.27$, correction (Hut, 1987) accounts for the difference in the reference scales of seawater (permil versus V-SMOW) and calcite (permil versus V-PDB).

In previous studies, we and others (Caley et al., 2014; Werner et al., 2016) computed the above $\delta^{18}O$ equilibrium value, averaged over time and the surface layer (typically the first 50 meters) to compare model results and measured $\delta^{18}O_c$ from planktonic foraminifers. In the following, we will refer to this method as the "old method", written formally as:

$$\delta^{18}O_{c,om}(x,y) = \frac{1}{n_t}\sum_{t=1}^{n_t}\frac{1}{n_z}\sum_{z=0}^{z_b}\delta^{18}O_{c,eq}\left(\delta^{18}O_{sw}(x,y,z,t), T_{sw}(x,y,z,t)\right) \tag{7}$$

where $n_t$ is the number of time steps, $n_z$ the number of vertical levels and $z_b$ the maximum depth.

The formalism used clearly expresses the fact that the old method is not species-specific nor season-specific since all time steps and vertical levels are averaged with the same weight. In contrast, the FAME method weighs the $\delta^{18}O_c$ both in time and in the horizontal and vertical space according to the population abundances using the foraminifer growth formula (1). We thus write:

$$\delta^{18}O_{c,fm}(x,y,k) = \frac{1}{n_t}\sum_{t=1}^{n_t}\frac{1}{n_z}\sum_{z=0}^{z_b(k)}\delta^{18}O_{c,eq}\left(\delta^{18}O_{sw}(x,y,z,t), T_{sw}(x,y,z,t)\right)\mu'\left(T(x,y,z,t),k\right) \tag{8}$$

where $z_b(k)$ is dependent on the species and constrained by core-top data (see below).





Using this set of equations, for any given seawater temperature and $\delta^{18}$O provided as a 4-dimensional field and a given species k, we compute this species $\delta^{18}$O$_c$ over x,y (latitude, longitude) coordinates.

It should be clearly understood that this approach is not able and does not attempt to determine the relative abundances of the
different species. Instead FAME provides a simplified approach to compute the $\delta^{18}$O$_c$ of a generic population of foraminifers if environmental conditions permit its growth. From a model – data perspective, this approach enables one to compute the calcite $\delta^{18}$O for a given species, were it to exist in the sedimentary record. Due to the limitations set by equation 3, no calcite isotopic content is computed if $\mu'$ is zero and hence these areas will be masked out in the following.

## 2.2   Reference datasets

In an attempt to validate the FAME approach, we apply its methodology to reference datasets, close to present-day observations. The first necessary step is the computation of a reference $\delta^{18}$O$_c$ field as obtained when forced by climatological data. For seawater temperature, we use the World Ocean Atlas 2013 (Locarnini et al., 2013) data at a monthly resolution. Considering that there is no equivalent seawater oxygen-18 gridded dataset available in the World Ocean Atlas fields and that the
existing GISS gridded dataset (LeGrande and Schmidt, 2006) presents large deviations with respect to the seawater oxygen-18 ($\delta^{18}$O$_{sw}$) raw data in numerous locations, we derived a $\delta^{18}$O$_{sw}$ dataset based on seawater salinity to $\delta^{18}$O$_{sw}$ relationships. This dataset is built in two steps: a) derivation of regional $\delta^{18}$O$_{sw}$ – salinity relationships from GISS $\delta^{18}$O$_{sw}$ and salinity (Schmidt et al., 1999) clustered by oceanic regions and b) computation of a $\delta^{18}$O$_{sw}$ field based on the World Ocean Atlas 2013 (Zweng et al., 2013) salinity fields. The resulting field is at the World Ocean Atlas spatial resolution and is used as reference seawater
oxygen-18 in the following. Details on the derivation of the $\delta^{18}$O$_{sw}$ dataset are given in Appendix A.

As an independent test of the FAME results, we use the planktonic $\delta^{18}$O$_c$ measurements from the MARGO Late Holocene dataset (Waelbroeck et al., 2005) restricted to high chronozone quality levels (i.e. levels 1 to 4). A few errors have been corrected in the published data set: these concern the suppression of 10 *Neogloboquadrina incompta* (or *N. pachyderma right*) data points
from the Nordic Seas where only *Neogloboquadrina pachyderma* should have been listed, and one outlier *N. pachyderma* value with no age control that was erroneously listed as having a level 4 chronozone quality. The corrected version of MARGO Late Holocene planktonic oxygen-18 data set is available as supplementary material.

As a result, the dataset used in the present study contains 248 values for *Neogloboquadrina pachyderma*, 128 values for *Globigerina bulloides*, 59 values for *Neogloboquadrina incompta*, 135 values for *Globigerinoides ruber* and 51 values for
*Globigerinoides sacculifer*.



## 2.3 Calculation of the best-fitting maximum depth per foraminifer species

In equation 8, the maximum depth of integration per foraminifer species, $z_b$(k), is a free parameter and needs to be determined. We have chosen to calculate it as the depth where the $\delta^{18}O_c$ simulated by FAME driven by the World Ocean Atlas 2013 temperature and derived seawater $\delta^{18}O$ datasets is closest on average to MARGO Late Holocene $\delta^{18}O_c$ data. To determine the optimal value of $z_b$(k), we repeated successive runs of FAME with values of $z_b$ ranging from 1,500 meters till the surface along the standard World Ocean Atlas vertical grid. The only difference between the different species at this stage are the species-specific terms in the equations presented and the each species data from the MARGO Late Holocene set. The results obtained through this optimization procedure are given in Table 1. The maximum depths of calcification derived this way are remarkably close to what is known from the ecology of *G. ruber*, *N. incompta*, *G. sacculifer* and *G. bulloides* (Berger, 1969; Bijma and Hemleben, 1994; Ortiz et al., 1995; Schiebel et al., 2002; Mortyn and Charles, 2003; Rebotim et al., 2017). Only in the case of *N. pachyderma*, the computed value of $z_b$ was much too deep (900 meters) with respect to what studies based on opening-closing plankton nets show. Also, plankton hauls studies have revealed that whereas *N. pachyderma* seems to grow at relatively shallow depth, i.e. where the chlorophyll maximum is found, a calcite crust is added between 50 and 250 m, which greatly increases its mass (Kohfeld et al., 1996; Simstich et al., 2003). As a consequence, the $\delta^{18}O$ of *N. pachyderma* collected in deep sediment traps and in surface sediment is systematically heavier than that of living non-encrusted *N. pachyderma*. To account for this effect we have added a 0.1 per mil "encrustation term" to our calculation of *N. pachyderma* calcite $\delta^{18}O$ weighted by that species culture-based growth rates.

The relatively deeper habitat depth derived for *G. sacculifer* versus *G. ruber* (maximum calcification depth estimates range from -200 to -75 m, best estimate = -100 m) could result from the increase in $\delta^{18}O$ due to accretion of gametogenic calcite (for a certain unknown fraction of the shell mass) or precipitation of its final sac-like chamber deeper in the water column. In addition, some of the *G. sacculifer* $\delta^{18}O$ data were obtained from deep Pacific cores in which dissolution might have induced a 0.2 to 0.5 permil enrichment in $\delta^{18}O$. Dissolution biases or contribution of a large anomalous signal to individual shells $\delta^{18}O_c$, as a terminal feature from deeper in the water column, may reconcile the discrepancy between apparent living depths recorded by water column sampling and sediment fossil assemblages.

## 2.4 Evaluation of the model performance

### 2.4.1 Error distribution

Since the depth parameter was constrained using the MARGO Late Holocene dataset by error minimization, it is not surprising that the errors obtained with FAME are very small in average for each species considered (Figure 2). The error distribution obtained with FAME is very similar to the one obtained with the simple surface equilibrium assumption for the two species closest to the surface (*G. ruber* and *N. incompta*). For deeper dwellers (*G. sacculifer*, *G. bulloides* & *N. pachyderma*) FAME results are better than those obtained with the old method, as expected since deeper layers in the ocean are accounted for.





**Table 1.** Maximum depth per species as computed from the optimization procedure. $z_b$ is the depth yielding the smallest difference to the Late Holocene MARGO data (Waelbroeck et al., 2005). We computed a confidence interval $\left[\sigma_{z_d}^{\uparrow}, \sigma_{z_d}^{\downarrow}\right]$ corresponding to a change of $\pm 0.1$ per mil in the mean error. The $\infty$ sign indicates that no value of $z_b$ within the range $[0, -1500]$ yields the desired $\pm 0.1$ per mil change.

| Species | $z_b$ (m) | $\left[\sigma_{z_d}^{\uparrow}, \sigma_{z_d}^{\downarrow}\right]$ (m) | Nb points |
|---|---|---|---|
| *G. ruber* | -10 | $]\infty, -30]$ | 130 |
| *N. incompta* | -65 | $[-35, -150]$ | 60 |
| *G. sacculifer* | -100 | $[-75, -200]$ | 46 |
| *G. bulloides* | -400 | $[-100, -\infty[$ | 123 |
| *N. pachyderma* | -550[1] | $[-275, -900]$ | 244 |

[1] an encrustation term of $+0.1$ per mil is taken into account in the case of
N. pachyderma (see text)

### 2.4.2 Geographical distribution

To further ascertain our methodology against the MARGO Late Holocene dataset we compare the zone of presence of each species predicted by FAME (grossly determined by $\mu'$) with the observed presence in the MARGO dataset (restricted to chronology quality values 1-4). As noted above, we cannot predict the relative abundance of each species. However, the

method determines the species absence or presence.

The results presented in Figure 3 show that, despite the exceptional simplicity of our approach, FAME predicts relatively well the spatial limits of the area occupied by each species. The two species whose presence distribution is best predicted are again *G. bulloides* and *N. pachyderma*, both showing a quite remarkable model-data match of the transition zones from presence to absence. *N. incompta* and *G. ruber* also show quite satisfactorily results, with only a few outliers in specific areas: FAME

computes too extended coverages of *N. incompta* in the Gulf of Guinea and of *G. ruber* along the coast of Namibia.

The computed spatial coverage of *G. sacculifer* is slightly too extended towards high northern and – possibly – southern latitudes. The very low number of high quality dated datapoints in the latter area prevents a definitive conclusion. Also, specific zones, consistent for several species, may be noted such as the Benguela upwelling regions where FAME fails to predict the

absence of *G. sacculifer* and *G. ruber*.

One possible explanation for this mismatch could be the impact of increased nutrient availability on observed abundances as a consequence of the upwelling systems, wheras nutrients are at present ignored in the FAME approach. Another possibility could be the quality of the vertical oceanic structure obtained from the World Ocean Atlas in those upwelling regions. Finally, it should be noted that our comparison ignores the natural interannual variability since we are using climatologies. The inter-

annual variability involves changes in the location of the fronts and currents and thus bears the potential of shifting the spatial boundaries between the different foraminifer species.

Further discussion of the abundance comparison including all datapoints from the MARGO Late Holocene dataset regardless





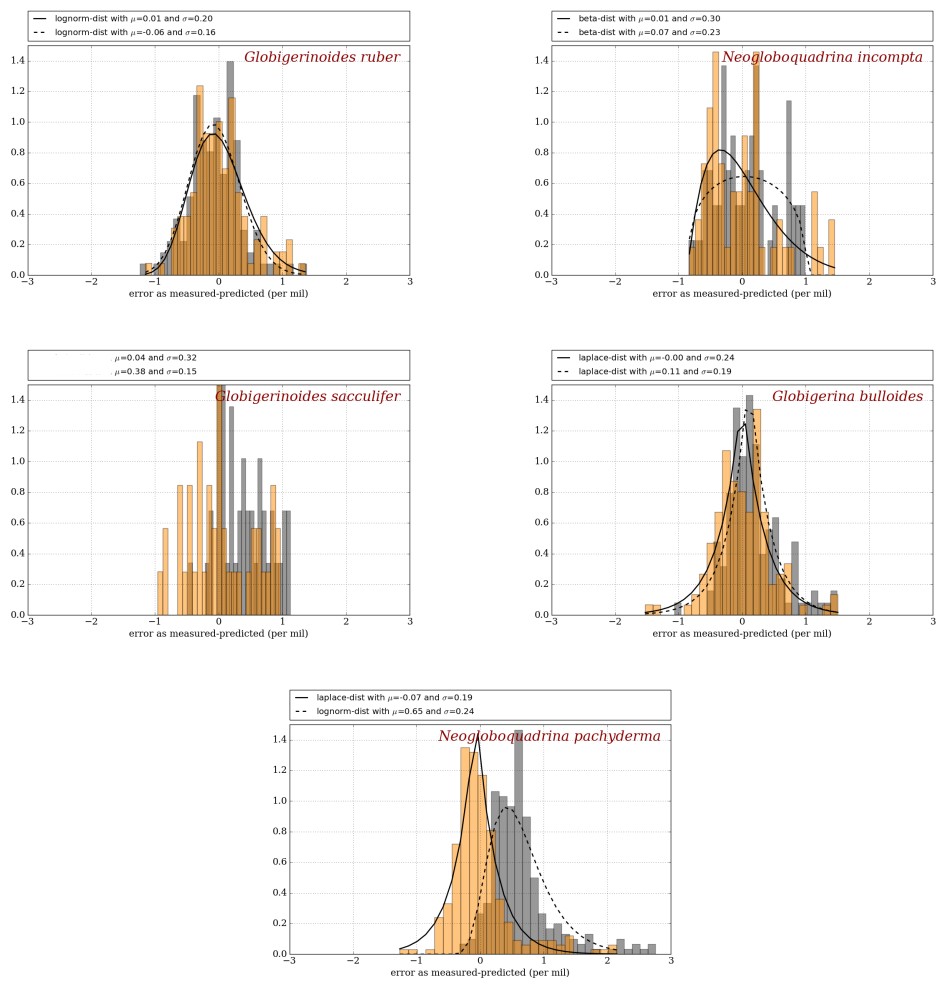

**Figure 2.** Error distribution for the "old method" (grey) and the "FAME method" (orange) using climatological datasets as compared to MARGO Late Holocene dataset (Waelbroeck et al., 2005). Best fitting distributions are calculated and plotted as a solid line for the "FAME method" and as a dashed line for the "old method", except for *G. sacculifer*, for which the small number of available data points yields a poor fit both for FAME and the old method. The mean and deviation are given for FAME and the old method at the top of each panels.

of the dating quality is given in Appendix B.

To further investigate the functioning of the FAME model, it is useful to consider the spatial distribution of the depth at which each species' growth is maximum. An example is given for the month of July in Figure 4. It clearly shows that even though the maximum depth allowed for each species is fixed through the $z_b(k)$ parameter, the predicted/computed calcification depth varies according to the location in the world ocean. Except for *G. ruber* which always calcifies in the topmost ocean layers, the depth of maximum growth exhibits large spatial variations, notably at the edge of the species' domains; in July this





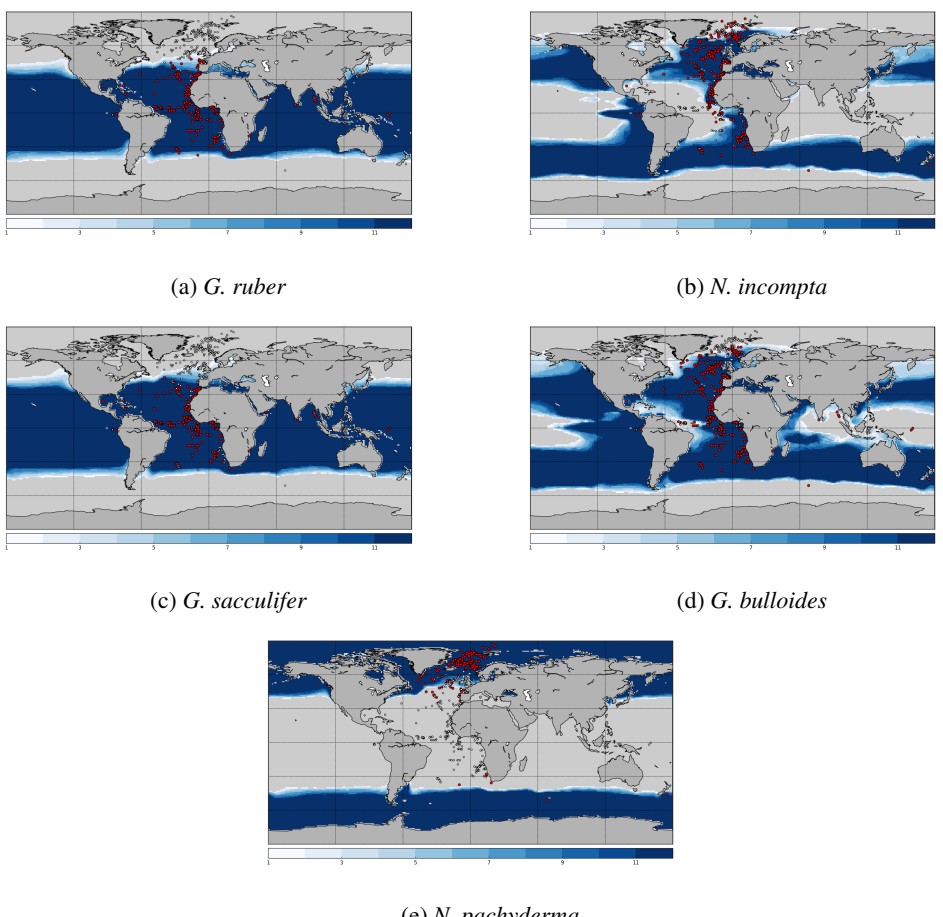

**Figure 3.** Model-data comparison of species abundances. Ocean regions where FAME predicts that the species is present at some time of the year ($\mu' > 0$) are plotted in blue, with shades of blue indicating the number of months of presence. Overlaid are the MARGO Late Holocene data: sites where the species' abundance is higher than or equal to 10% are shown by red dots while the other sites are marked by white dots.

is particularly marked in the case of *G. bulloides* and *N. pachyderma* (Figure 4d and 4e).

Likewise, it is useful to consider the seasonal variations in the depth of maximum growth for a given species. We propose to highlight this aspect for the two species that show the largest variations: *N. pachyderma* and *G. bulloides* at two extreme months (January and July) (Figure 5). For both species, the area of computed non-zero contribution varies along the year, with an expansion (reduction) of the area occupied by *N. pachyderma* in the northern hemisphere in January (July), while the regions occupied by *G. bulloides* shift towards higher (lower) latitudes in the northern hemisphere in July (January). These seasonal changes are a direct response of these species' preferred habitat to temperature. FAME thus mechanistically predicts the adaptation of planktonic foraminifer depth habitat to maintain optimal living conditions. For instance, Figure 5b and 5d clearly show that *G. bulloides* is predicted to dwell deeper at low latitudes when surface temperature rises above its preferred





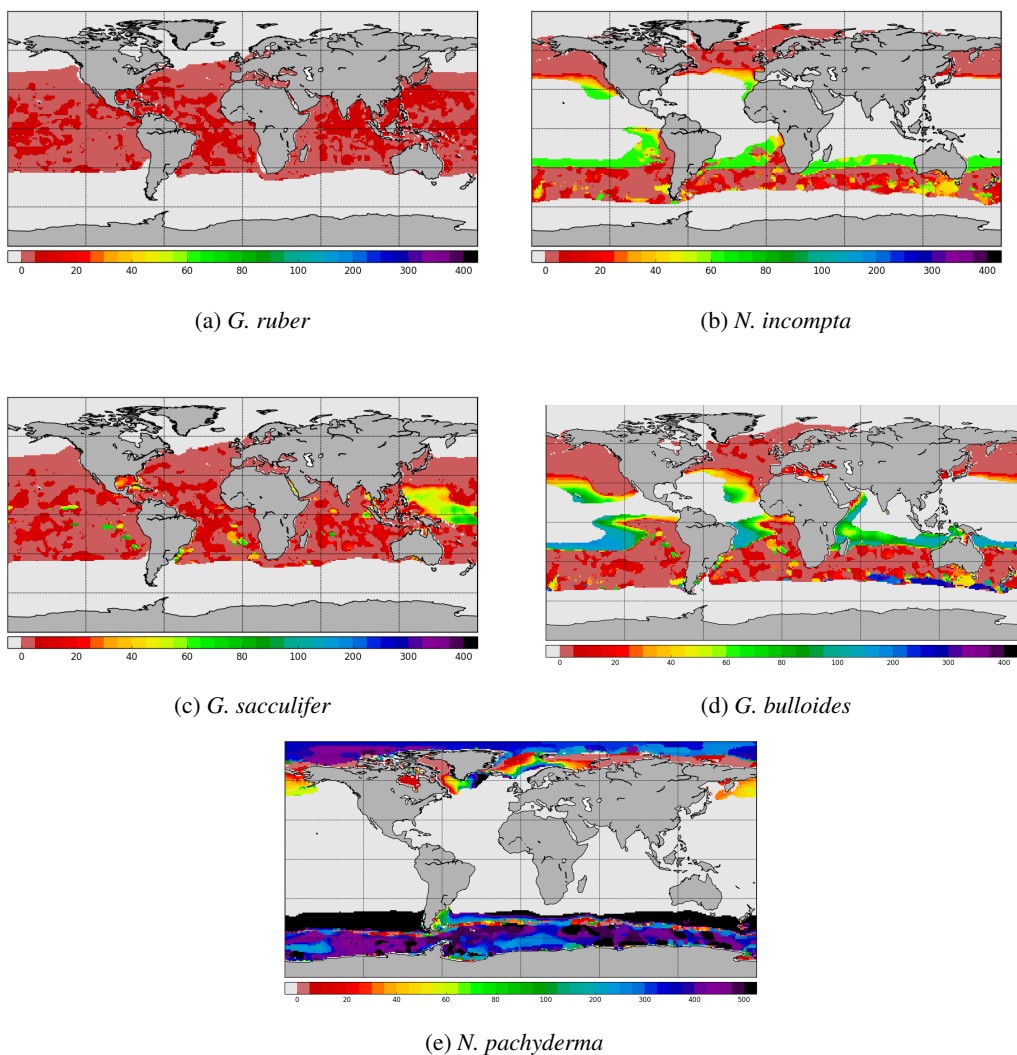

(a) *G. ruber*

(b) *N. incompta*

(c) *G. sacculifer*

(d) *G. bulloides*

(e) *N. pachyderma*

**Figure 4.** Depth of maximum growth for the species considered for the month of July. The color scale shows the depth in meters. Oceanic areas left in white correspond to areas where growth rates are below the threshold defined in equation (3).

temperature range. Similarly, Figure 5b and 5d show that *G. bulloides* is present at higher northern latitudes in July than in January, so that the growing season actually tracks the species preferred living conditions, as observed (Jonkers and Kucera, 2015).

### 2.4.3 Effect of a large climatic change on the computed oxygen-18 content of the calcite

5  Though FAME gives realistic results when forced by atlas data, it is mostly designed to retrieve the species specific effect of climate change on the recorded $\delta^{18}O_c$. To highlight the effect of seasonal and vertical weighting of the $\delta^{18}O_c$ signal computed



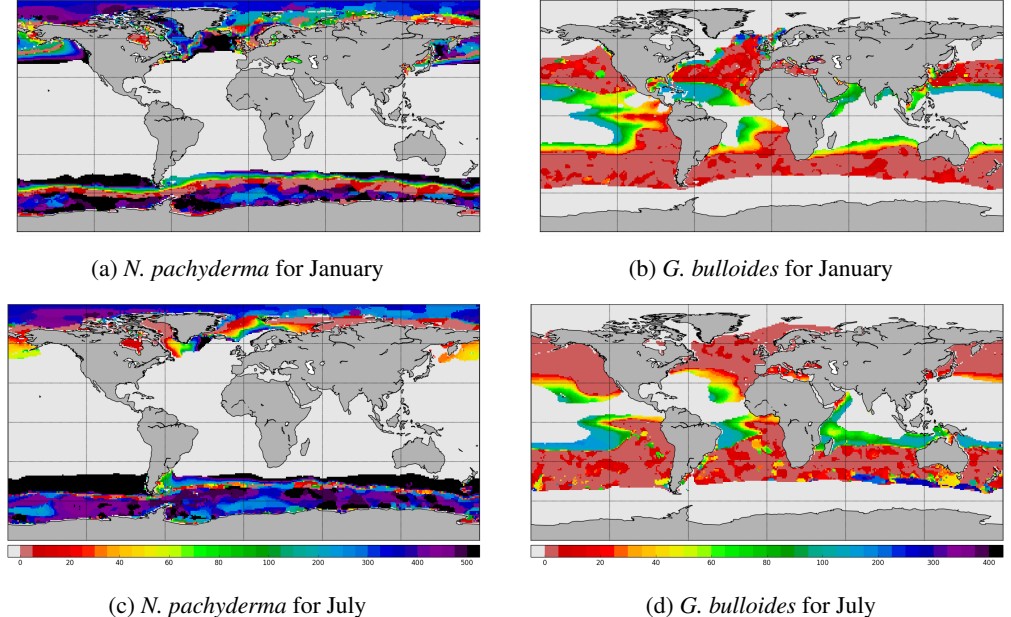

(a) *N. pachyderma* for January        (b) *G. bulloides* for January

(c) *N. pachyderma* for July        (d) *G. bulloides* for July

**Figure 5.** Comparison of the depth of maximum growth for N. pachyderma and G. bulloides for January and July. Color scale as in Figure 3.

by FAME, we have performed a simplified experiment showing the effect of a change in the foraminifers living conditions on their $\delta^{18}O_c$ signal.

To simulate a change in climatic conditions, we apply a homogeneous $4°C$ decrease to the WOA13 sea temperature dataset and compute the anomaly in $\delta^{18}O_c$ between that new cold state and the original one for each species as well as for the surface equilibrium approach (Figure 6). This anomaly is noted $\Delta^{18}O_c$ in what follows.

Applying a spatially homogeneous temperature change should result is a quasi-homogeneous temperature change in the equilibrium calcite $\Delta^{18}O_c$, following equation 6. It is indeed what is obtained in panel (e) of figure 6, with $\Delta^{18}O_c$ values between 0.8 in the tropics to 1 per mil at high latitudes. When applying the FAME equations, we obtain large spatial variations in $\Delta^{18}O_c$ with values down to -0.75 and up to 1 per mil. All species share a common pattern of lower $\Delta^{18}O_c$ at the border of their living domain and close to equilibrium values at the center of their living domain. More specifically, the smallest differences to the equilibrium are recorded by *N. pachyderma* and the largest, negative, differences are computed for *G. bulloides*. The species with the smallest vertical living range, *G. ruber*, has the most homogeneous distribution. The range of values (minimum to maximum) is always close to one per mil with the exception of *G. bulloides* that presents a total range of 1.6 per mil. This large range of $\Delta^{18}O_c$ for *G. bulloides* is a consequence of its growth over a large range of temperatures (equation 1). In general, the maximum simulated $\Delta^{18}O_c$ are systematically 0.1 to 0.2 per mil lower than the equilibrium value.



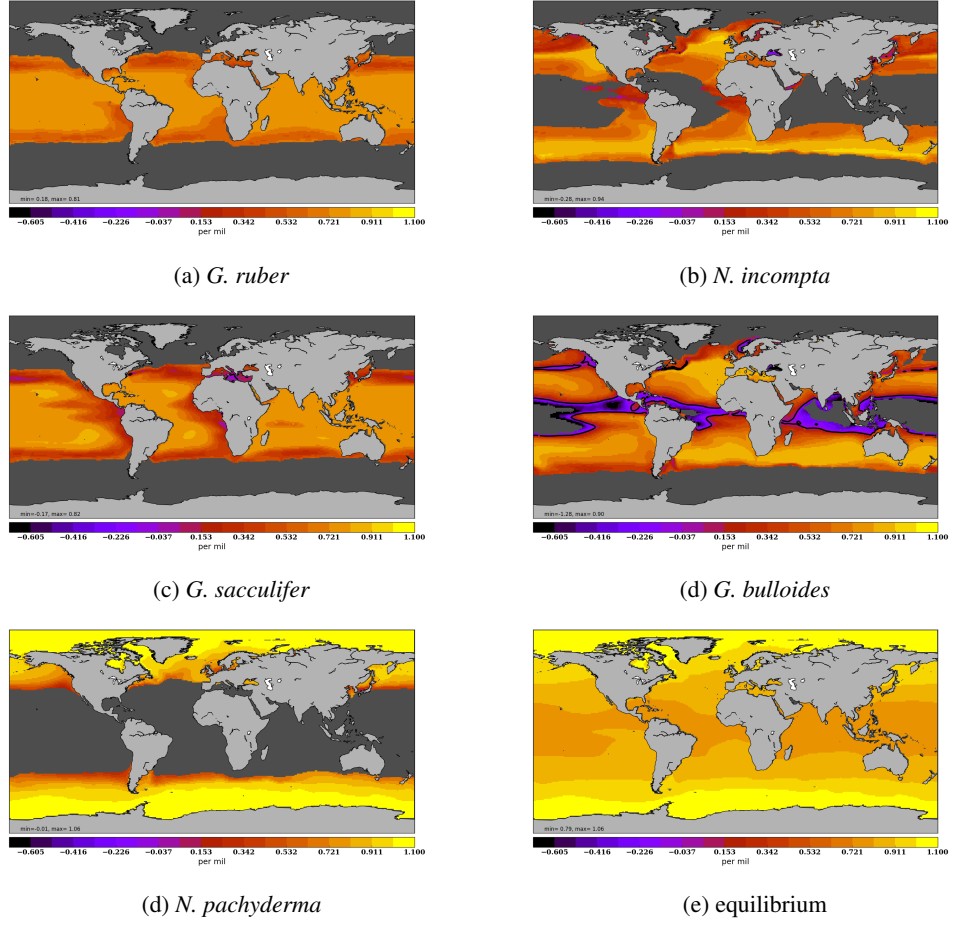

**Figure 6.** $\delta^{18}O_c$ response to a horizontally and vertically homogeneous $4°C$ cooling applied to the WOA13 dataset, $\Delta^{18}O_c$. Results are expressed in per mil for each species (panels (a-d)) and for the equilibrium surface calcite approach.

This simple scenario, though unrealistic with respect to actual climatic applications, shows the potential of FAME to unravel the climatic signal embedded in multi-species isotopic records and thus opens the door to transient climate – data intercomparison where the species' specific behaviour is taken into account.

## 3 Summary and conclusions

We developed the FAME (Foraminifers As Modeled Entities) module to account for planktonic foraminifer species-specific habitat when computing their calcite oxygen-18. In contrast to models predicting the abundance of planktonic foraminifers, FAME only aims at predicting the presence or absence of a given species and its oxygen-18 value. FAME is only forced by





hydrographic data, and uses a very limited number of parameters, almost all derived from culture experiments. Taken together, these characteristics make FAME a uniquely simple and robust model predicting changes in the oxygen-18 of commonly used planktonic foraminifer species in response to changing climatic conditions. FAME performance is evaluated using MARGO Late Holocene planktonic foraminifer $\delta^{18}O_c$ and abundances data sets. We show that FAME predicts remarkably well the

5 presence/absence of *G. ruber*, *N. incompta*, *N. pachyderma* and *G. bulloides* over most of the world ocean, while yielding a slightly less good prediction of *G. sacculifer* presence/absence. Investigating the simulated seasonal pattern, we show that the predicted growing season and habitat depth track the species preferred living conditions, as observed in plankton hauls and sediment trap data. Finally, the application of FAME to a simple cooling scenario demonstrates that computed changes in species-specific $\delta^{18}O_c$ are much more spatially variable than the computed change in equilibrium surface calcite. Coupling

the FAME module to isotope-enabled climate models makes it possible for the first time to extract the climatic information contained in isotopic time series measured on different planktonic species at the same location. This opens the possibility to better reconstruct the evolution of the upper water column structure than ever before, notably over climate transitions.

## 4  Code availability

The FAME module has been developped in python language version 3 and tested under version 3.5.1. The code is made
available under the GNU General Public License https://www.gnu.org/licenses/gpl.html and is uploaded as a supplement of this manuscript.

## 5  Data availability

The World Ocean Atlas datasets used are available to all users directly from the provider. Derivation of the a reference $\delta^{18}O_{sw}$ dataset is detailed in the Appendix A. The masks file used in the latter procedure is provided as a supplement to the manuscript.

## 20  Appendix A:  Derivation of the a reference $\delta^{18}O_{sw}$ dataset

We constructed our reference $\delta^{18}O_{sw}$ dataset at World Ocean Atlas standard resolution (1° grid) through a three steps methodology: a) construction of an appropriate basin mask to allow clustering the GISS global dataset regionally b) derivation of $\delta^{18}O_{sw}$ – salinity relationships for each of these basins and c) use of these relationships to obtain a $\delta^{18}O_{sw}$ field at WOA spatial resolution.

## 25  A1  Construction of the basin masks

Our native resolution being the 1° regular grid of the World Ocean Atlas, we first retreived the available basin mask file on that grid from the NOAA website https://www.nodc.noaa.gov/OC5/woa13/masks13.html and converted it to a netCDF format file (http://www.unidata.ucar.edu/software/netcdf). The basins defined in the WOA base mask did not perfectly fit our purpose, we





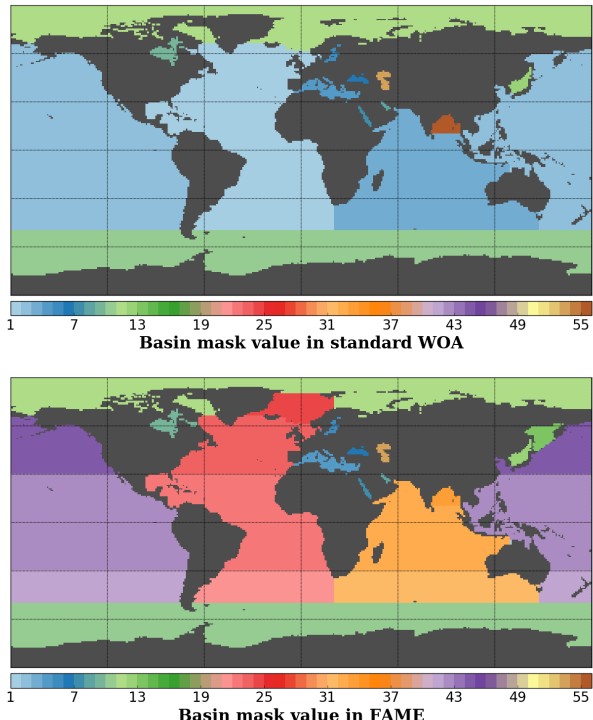

**Figure A1.** Basin masks as defined in the WOA standard mask file (above) and in FAME, on the same $1°$ resolution grid. Values corresponds to the basins defined in Table A1.

hence modified the masks to isolate some particular regions where the $\delta^{18}O_{sw}$ and salinities are specific (e.g. Sea of Okhotsk) or merge some regions of the WOA mask into larger ensembles (e.g. Hudson Bay). A summary of the basins in the original file and in ours is given in Table A1. The Pacific and Atlantic oceans were split into South, North and Tropical parts, based on boundaries at $30°$ North and South respectively. The Indian Ocean has been only split in two: North and South using the $30°$ South boundary. The Bay of Bengal has been kept a separated basin as in the original file. The GIN Seas were made a separated basin from the Arctic Ocean, using the boundaries at $80°$ North and at $20°$ East. We also extended the Hudson Bay mask area to include the Hudson Strait and Ungaya Bay, since these do not represent the same water mass properties as the North Atlantic Ocean. The limit used is $-64.5°$ West, corresponding to the southern tip of Resolution Island on the grid given. Finally, the same procedure was applied to define the Okhostk Sea, using the official definition of the International Hydrographic Organization (IHO SP-23). The results of this whole procedure is shown in Figure A1. In Table A1, some values are annotated with a "*" to highlight basins having the same value in FAME as in the standard WOA but covering a different area: the Arctic Ocean from which the GIN seas have been taken out in FAME, Hudson Bay which covers a part of the former Atlantic basin of the WOA given its afore mentioned expansion to the Hudson Strait and Ungaya Bay.

The netCDF data file resulting from this procedure is provided as a supplement to the manuscript.





**Table A1.** Comparative list of basin masks in WOA and FAME. The "value" field provides the integer value used in the netCDF file to specify the respective basin on the WOA grid.

| Basin name | WOA value | FAME value |
| --- | --- | --- |
| Atlantic Ocean | 1 | |
| South Atlantic O. | | 21 |
| Tropical Atlantic O. | | 22 |
| North Atlantic O. | | 23 |
| GIN Seas | | 24 |
| Pacific Ocean | 2 | |
| South Pacific O. | | 41 |
| Tropical Pacific O. | | 42 |
| North Pacific O. | | 44 |
| Indian Ocean | 3 | |
| South Indian O. | | 31 |
| North Indian O. | | 32 |
| Mediterranean Sea | 4 | 4 |
| Baltic Sea | 5 | 5 |
| Black Sea | 6 | 6 |
| Red Sea | 7 | 7 |
| Persian Gulf | 8 | 8 |
| Hudson Bay | 9 | 9* |
| Southern Ocean | 10 | 10 |
| Arctic Ocean | 11 | 11* |
| Sea of Japan | 12 | 12 |
| Okhotsk Sea | | 13 |
| Caspian Sea | 53 | 53 |
| Bay of Bengal | 56 | 33 |

\* highlights the regions where FAME and WOA regions do not cover
the same area (see text))

## A2   Computation of the $\delta^{18}O_{sw}$ – salinity relationships

The basins defined in the previous section are then used to cluster the raw data, $\delta^{18}O_{sw}$ and salinity, of the GISS database (Schmidt et al., 1999) in the respective basins. Furthermore, only data locations where both $\delta^{18}O_{sw}$ and salinity are given in the original database for depths less than 200 meters are retained under the additional constraint that depth of the ocean should be more than 175 meters. The latter to ensure that the values are representative of high seas values and not to coastal areas, possibly under fluvial influence. Additionally, all values below 5 per mil in salinity are ignored for all basins. Lastly, for two

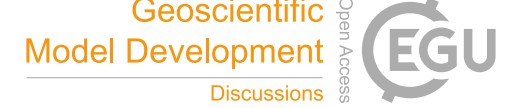



**Table A2.** Values obtained for the $\delta^{18}O_{sw}$ – salinity relationships

| Basin name | slope | intercept | $R^2$ | Nb. points |
|---|---|---|---|---|
| South Atlantic O. | 0.52 | -18.02 | 0.95 | 55 |
| Tropical Atlantic O. | 0.25 | -8.19 | 0.60 | 241 |
| North Atlantic O. | 0.51 | -17.65 | 0.67 | 738 |
| GIN Seas | 0.69 | -23.72 | 0.73 | 1471 |
| South Pacific O. | 0.42 | -14.45 | 0.92 | 19 |
| Tropical Pacific O. | 0.31 | -10.36 | 0.76 | 417 |
| North Pacific O. | 0.43 | -14.69 | 0.92 | 333 |
| South Indian O. | 0.53 | -18.39 | 0.89 | 255 |
| North Indian O. | 0.10 | -2.88 | 0.40 | 466 |
| Mediterranean Sea | 0.27 | -8.98 | 0.61 | 196 |
| Baltic Sea | 0.34 | -9.14 | 0.63 | 21 |
| Black Sea | 0.28 | -6.77 | 0.06 | 18 |
| Red Sea | 0.28 | -9.61 | 0.97 | 16 |
| Hudson Bay | 0.40 | -15.33 | 0.47 | 286 |
| Southern Ocean | 0.42 | -14.84 | 0.80 | 1005 |
| Arctic Ocean | 0.54 | -18.82 | 0.72 | 2932 |
| Sea of Japan | 0.36 | -12.83 | 0.94 | 45 |
| Okhotsk Sea | 0.42 | -14.46 | 0.93 | 453 |
| Bay of Bengal | 0.24 | -7.9 | 0.40 | 131 |

basins (North Atlantic and Bay of Bengal), the existence of two different slopes where only one corresponds to open ocean conditions render necessary the addition of one additional condition to keep only the latter. We thus added a limit at 27 per mil in salinity for those two basins.

5    The resulting slopes, intercept and correlation coefficients are given in the Table A2. Using those relationships, we further compute the $\delta^{18}O_{sw}$ in the WOA geographical grid from the WOA salinity fields.

## Appendix B:  Further discussion of predicted and observed planktonic foraminifer abundances

In the main text, we have only compared the results of FAME to the datapoints in the MARGO Late Holocene database that
10    were characterized by high chronological control quality. Since this drastically restricts the geographical extent covered by MARGO data and in the interest of completeness, we propose here a short discussion based on all points of the MARGO Late





Holocene database, regardless of their chronological control quality. The interest of figure A2 is to provide some information in the Southern, Pacific and Indian Ocean regions that are largely void of points in the previous figure. While the bulk of the conclusions given in the main text is unchanged by this new comparison, we may highlight the following.

The unsorted distribution for *G. ruber* is not very different from the one described above, albeit with a good definition of the Southern Ocean abundance limit where FAME results are in good accordance with MARGO. Also, one may note a series of points without the presence of *G. ruber* in the equatorial Pacific in MARGO, an aspect which is not predicted by FAME. However, these points are mingled with points with *G. ruber* presence in the MARGO database, indicating they could be an artifact resulting from the presence of older sedimentary material in the unsorted MARGO database ; it is thus difficult to draw
a firm conclusion.

Regarding *N. incompta*, the picture is pretty much the same as descibed in the main text to the exception of a number of mismatching sites in the tropical an mid latitudes in all southern ocean basins (Pacific, Indian and Atlantic).

The distribution for *G. sacculifer* shows a clear latitudinal mismatch of the limit of presence/absence when comparing the FAME results to the unsorted MARGO dataset. It seems obvious that the latitudinal spread of *G. ruber* in FAME should be
considered as too extended in the mid to high latitudes in both hemispheres.

The joint comparison of unsorted *G. ruber* and *G. sacculifer* distributions points to the existence of consistent zones where FAME does not predict the absence of those two species. This was noted earlier for the Benguela upwelling region. It is also visible here for the Peru-Chili upwelling and the eastern Equatorial Pacific. All these zones correspond to upwelling regions (e.g. Mackas et al., 2006) and are characterized by strong constrasts in surface water properties with respect to the surrounding
regions, large interannual and intra-seasonal variability, and high phytoplancton production. The existence of this consistent pattern in upwelling regions in the unsorted database confirms that *G. ruber* and *G. sacculifer* distributions are not well simulated in upwelling regions, either because nutrients are presently not accounted for in FAME, or because the increased nutrient availability and/or the vertical structure of oceanic physical properties is not faithfully depicted in the 1° resolution WOA13 dataset we used in input.

The unsorted distribution for *G. bulloides* still presents an excellent match for the limits, but some discrepancies in the equatorial and tropical latitudes, albeit MARGO unsorted data do not present a large regional consistency outside the northern coast of Brazil (where FAME also predicts the absence of *G. bulloides*).

Outside some minor mismatches in the southern Indian Ocean, the conclusions for *N. pachyderma* are also largely unaffected by the use of all the points from the MARGO database.


To conclude, the use of all the datapoints regardless of the quality of the chronological control in the MARGO Late Holocene database does not add much new information. Especially since the datapoints should be considered with caution as they could correspond to a different climate regime than the Late Holocene.





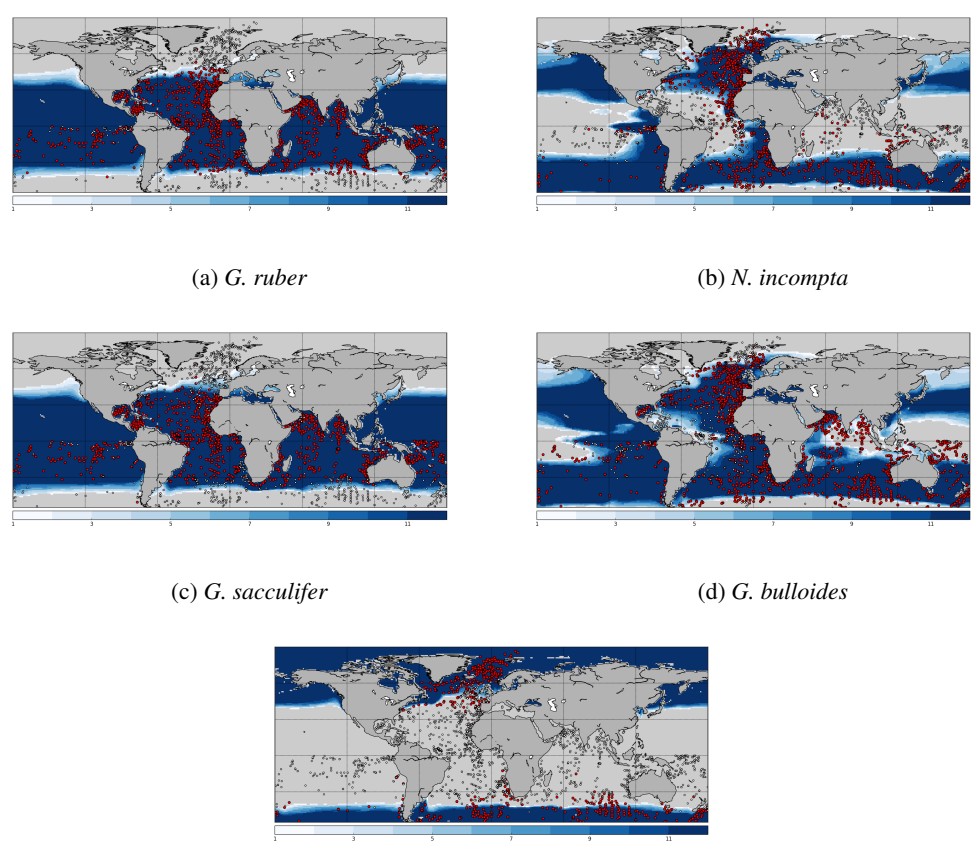

(a) *G. ruber*  (b) *N. incompta*

(c) *G. sacculifer*  (d) *G. bulloides*

(e) *N. pachyderma*

**Figure A2.** Model-data comparison of species abundances. Ocean regions where FAME predicts that the species is present at some time of the year ($\mu' > 0$) are plotted in blue, with shades of blue indicating the number of months of presence. Overlaid are the MARGO Late Holocene data: sites where the species' abundance is higher than or equal to 10% are shown by red dots while the other sites are marked by white dots.

*Competing interests.* The authors declared no competing interests.

*Acknowledgements.* This is a contribution to the ACCLIMATE ERC project. The research leading to these results has received funding from the European Research Council under the European Union's Seventh Framework Programme (FP7/2007-2013 Grant agreement n° 339108). We thank L. Jonkers and M. Kucera for fruitful discussions on earlier versions of this work.



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
