# Peer review of "FAME (v1.0): a simple module to simulate the effect of planktonic foraminifer species-specific habitat on their oxygen isotopic content"

_Geoscientific Model Development, 2017_

## Referee Comment (RC1) · Anonymous Referee #1 · 16 Jan 2018

The manuscript by D. Roche et al, proposes a new module to simulate the effect of foram species-specific habitat (namely depth and season) on their isotopic content. This short and well-structured manuscript builds on the FORAMCLIM model to draw conclusions on the specific living depth of foraminifera. I greatly appreciated the fact that the code is open source, which will be of help not only for modellers but also for the paleo and modern foram community.

The manuscript posits that the growth and habitat of planktonic foraminifera can be simply described using a set of parameters, derived from culture experiments, which are mostly dependent of the temperature. This is based on the Lombard et al, 2009

FORAMCLIM model. This model is compared to the Late Holocene MARGO data-set, in order to have a first order idea of the distribution of planktonic foraminifera, and through the coupling with the d18O module computation from salinity-regressions, the range of d18O one could expect for a significant cooling ($\Delta T=4°C$).

This is a stimulating contribution which from a micropaleontologist point of view raises a few questions, and two main ones.

My first concern in the manuscript is the model-data comparison: there are some visual comparisons by overlying the percentage of foraminifera species to the presence at some time of the year, with an ad-hoc threshold at 10%. As the focus is based on the oxygen isotopes, I wonder why the authors use the species distribution for testing this model which has been already validated by Lombard on plankton nets and surface sediments (just using the surface temperature). The issue here is to find an independent data-set to validate their isotopic model. I wonder if some stable isotope sediment trap data could not be a better benchmark to validate the model, rather than the comparison with surface sediments. There are some time series in the South China Sea (Lin et al, 2011); in the Gulf of Mexico (Thiraulamai et al, 2015), and the works lead by R. Thunell among others.

A second concern is the propagation of errors throughout the model which is not properly dealt. As the model picks a best fit for the response of the growth of foraminifera to temperature, and that in the code (and in the original paper), the uncertainties are given, it would be useful to propagate the errors of the response (growth) of foraminifera to temperature. It would be extremely useful for the community, as it would give the reader a sense of the sensitivity of the models. This point is also detailed below.

Technical comments :

[l. 18 page 1]: Expand the connection between the stratified nets and the isotopic derived values of Emiliani more in details. In Jones, there is no reference whatsoever to any isotopic analyses. You are making the connexion, but this was not put forward

by Jones.

[page 2]: "It is this tempting to make one additional step": I do not understand this statement : It is a fact that oxygen isotopes have been implemented in models, but yet, this is not the topic of the paper as the d18O values of seawater are computed from empirical basin correlations between d18Osw and salinity, not from water isotopes enabled models. I would recommend to move this sentence in the perspectives, as it is misleading here and one quick reader might think that isotope models were used.

[equation 1, page 3]: I would add here a reference to the original work (Kooijman 2000) which formalized this equation as referred in Lombard et al (2009).

[line 5, page 4]: The authors do use a Tl of 280 for G. bulloides instead of 281.1. This shows that the model is extremely sensitive to a minor change in Tl : could you please elaborate a bit on the reason on this 1.1°K shift ? Did you perform some sensitivity analyses to reach this temperature ? This is appealing because the overall inferred isotopic equilibrium depth calculated for this species is off the charts (see point below).

[Figure 1 page 4]: Fig 1 - I would add the original data as in this figure we lose the range of amplitude observed in cultures

[Figure 1 page 4]: Add reference to Lombard et al., 2009 in the figure caption

[page 7] Calculation of the best-fitting maximum depth: • What is the rationale for assuming that the Late Holocene equilibrium isotopic value value would be the maximum depth in the model ? Do you imply that the isotopic signature of foraminifera is biased toward the maximum calcification depth ? • The range of the depths calculated by the model are very deep compared to observed living depth. The most extreme case is G. bulloides : if one uses the last textbook written by R. Schiebel & Hemleben (Modern planktonic foraminifera, 2017) "Ecology: Globigerina bulloides mainly dwells above the thermocline within the upper 60 m of the water column, and is a non-symbiotic species". The ecology of this species is extremely problematic, and

likely due to a combination of multiple cryptic species (eg Morard et al), I would tend to think that the cultures did not catch the overall variability in the dataset. • I do not understand how does G. ruber has a living range reaching +∞. It would be extremely useful to have a figure putting into context the ranges (by comparing with Rebotim et al, for example event though this is a single figure).

[lines 18-24 - page 7]: I do agree that those two effects (gametogenic calcite and dissolution) can somehow impact the signature of d18Oc in G. sacculifer. Yet, as G. sacculifer is bearing symbionts, it does have to live in the euphotic zone, which is not the case in the model. I suggest that the authors make a more solid case by removing the deep Pacific sites that they supposed to be influenced by the dissolution to check whether the origin of this deep signature is indeed mostly gametogenic.

[lines 28-30 - page 7]: As the error scheme does not include the error linked to the calibration of the FORAMCLIM model. It would be extremely interesting to have an idea of the sensitivity of the FAME model to the max/min range observed in the data set.

[line 31 - page 7]: I disagree with the statement that G. sacculifer and G. bulloides can be called "deeper dwellers". The output of the model does rank them as deeper dwellers, but out at sea, they do live mostly in surface to subsurface layers of the ocean (see for example Schiebel and Hemleben, 2017).

[Table1 - page 8]: The range is definitely too deep for G. bulloides (ibid.)

[Figure 3 - page 10]: I do not really understand what the figure does show : a percentage is highly depending of other species percentages – see my main comment #2. What is the rationale for the cutoff at 10% ? I do not see a physical nor biological rationale for this cutoff. I am wondering if the spatial coverage in the Indo Pacific Ocean is good enough to be included in the analysis as most core tops come the Atlantic Ocean.

[Figure 4 - page 11]: Consider changing the color scheme- rainbow does not give the

best rendition.

[page 12]: Add a latitudinal/depth plot, it would be more easy to read.

Please also note the supplement to this comment:
https://www.geosci-model-dev-discuss.net/gmd-2017-251/gmd-2017-251-RC1-supplement.pdf

---

## Referee Comment (RC2) · Anonymous Referee #2 · 18 Apr 2018

The manuscript by Roche et al. summarises a noble and interesting attempt to improve our understanding of foram-based oxygen isotope data. The authors present a module ('FAME' – Foraminifers As Modeled Entities) they developed in order to predict changes in the oxygen isotope composition of the tests of different foraminifera species in response to changing climatic conditions. The model is forced by hydrographic data alone and incorporates a limited number of species-specific parameters, based on culture experiments, for each of five foraminifera species to describe their growth and habitat. Essentially, the model attempts to account for the effect of foraminifera depth habitat on their oxygen isotope composition, and to predict their oxygen isotope composition accordingly, as well as their presence/absence. To test their model they apply

its methodology to reference datasets, namely the MARGO Late Holocene dataset. It is an interesting and concise presentation of their work and well-structured. I believe it will greatly contribute to research within the foraminifera and palaeoclimate community.

When such models are developed it's important to have some measure of their sensitivity. For that reason, I believe that error propagation in the model should be addressed given that several of the input parameters have errors associated with them.

Secondly, and this may sound pedantic but the authors may consider changing Globigineroides sacculifer to Trilobatus sacculifer as per its genus reassignment by Spezzaferri et al. (2015). I will leave this to the authors' discretion as there are arguments for retaining G. sacculifer given that this is still the most commonly used name for this species. However, over time this will obviously change and the authors may want to introduce the new (and more taxonomically up-to-date) name.

In terms of convention, there are several instances where the author refers to oxygen isotopes incorrectly. For example, p1, line 16, the authors describe the 'oxygen-18 value', or in line 21, 'calcite oxygen-18', or elsewhere as 'species' oxygen-18' (e.g. p3, line 1). This is very pernickety but there are quite strict guidelines for isotopic notation. I suggest the authors double check their usage and perhaps refer to ratios rather than oxygen-18 content/signal as it's more in line with the literature.

On page 7, line 16-17, you describe how you used a 0.1 per mil 'encrustation term'. Could you possibly elaborate as to where that value came from? It would make it easier for the reader as it seems a little arbitrary at present.

Also, the authors should mention wherever necessary that species with symbionts e.g. G. sacculifer (T. sacculifer) cannot live at depths greater than the photic zone, as is hinted at on page 7, line 20.

Some more specific comments: Page 1, line 17. Perhaps use 'reflected' rather than 'favoured'. Line 20. Use 'throughout the year' rather than 'along the year' as this

makes more grammatical sense. Page 2, line 7. I would consider adding a few more references here as several other studies have been done looking at carbonate ion and symbiotic effects. Pearson et al. (2012) gives a good summary of work up to that point. Line 26. Change 'being' to 'to be' Page 3, line 19. Italicise N. pachyderma. Page 5, line 21. Change 'weighs' to 'weight'? Page 8, line 2. Use a different word to 'ascertain' as this doesn't make sense in the context.\
* * *

---

## Author Comment (AC1) · 27 Jun 2018

*Response to Anonymous Referee #1*

*NOTA: The initial reviewer comments are in italic, our answers are in bold, action taken in the revised version of the manuscript are underlined.*

*The manuscript by D. Roche et al, proposes a new module to simulate the effect of foram species-specific habitat (namely depth and season) on their isotopic content. This short and well-structured manuscript builds on the FORAMCLIM model to draw conclusions on the specific living depth of foraminifera. I greatly appreciated the fact that the code is open source, which will be of help not only for modellers but also for the paleo and modern foram community.*

*The manuscript posits that the growth and habitat of planktonic foraminifera can be simply described using a set of parameters, derived from culture experiments, which are mostly dependent of the temperature. This is based on the Lombard et al, 2009 FORAMCLIM model. This model is compared to the Late Holocene MARGO data-set, in order to have a first order idea of the distribution of planktonic foraminifera, and through the coupling with the d18O module computation from salinity-regressions, the range of d18O one could expect for a significant cooling (ΔT=4∘C).*

*This is a stimulating contribution which from a micropaleontologist point of view raises a few questions, and two main ones.*

**We thank the reviewer for this positive view on our approach and on the manuscript. Responses to individual comments are dealt with below.**

*My first concern in the manuscript is the model-data comparison: there are some visual comparisons by overlying the percentage of foraminifera species to the presence at some time of the year, with an ad-hoc threshold at 10%. As the focus is based on the oxygen isotopes, I wonder why the authors use the species distribution for testing this model which has been already validated by Lombard on plankton nets and surface sediments (just using the surface temperature).*

**FORAMCLIM, the model validated by Lombard on plankton nets and surface sediments, computes foraminifer abundances based on food availability, light and temperature (Lombard et al., 2011). In contrast, FAME does not compute species abundances, but only growth rates as a function of temperature. Consequently, the validation steps we present for FAME have not been previously published. We would also like to stress that unlike Lombard's validation, we do not compare simulated against observed abundances, but simulated absence/presence against observed absence/presence. For these two reasons, the test we performed is different from the validation of the FORAMCLIM model published in Lombard et al. (2011).**

**We used a threshold at 10% to account for the census counts uncertainty is thus used as an indication of the absence / presence only. The reviewer is**

perfectly right that this 10% threshold is ad hoc. We have thus removed it from the revised version of the manuscript, where we present the MARGO data with colored abundances from 0 to 100%.

Action taken:
We have modified the previous figure 3 and A2 to comply with the reviewer's suggestion of not using an ad hoc threshold of 10%. The accompanying text as been modified accordingly.

*The issue here is to find an independent data-set to validate their isotopic model. I wonder if some stable isotope sediment trap data could not be a better benchmark to validate the model, rather than the comparison with surface sediments. There are some time series in the South China Sea (Lin et al, 2011); in the Gulf of Mexico (Thiraulamai et al, 2015), and the works lead by R. Thunell among others.*

The FAME module was developed in order to be coupled to the iLOVECLIM climate model and other climate models to simulate the isotopic signal of foraminifera in marine sediment cores. In doing so, our purpose is to enable model-data comparison via the simulation of planktonic foraminifer isotopic signals that can be directly compared to data derived from sediment cores. It is thus more relevant to validate the model against data from the sediment than from sediment traps or plankton tows. For the latter, the short cup window (~7-8 days), which is shorter than the duration of the foraminiferal life cycle in many cases, can add further sources of error when integrating the climate signal"
An additional problem is the availability of multiple species from sediment trap data: it is a real issue to find all needed species on a common sediment-trap data. On the other hand, the MARGO database yields directly a product which is very much comparable since it was compiled with inter-species comparison in mind.

*A second concern is the propagation of errors throughout the model which is not properly dealt. As the model picks a best fit for the response of the growth of foraminifera to temperature, and that in the code (and in the original paper), the uncertainties are given, it would be useful to propagate the errors of the response (growth) of foraminifera to temperature. It would be extremely useful for the community, as it would give the reader a sense of the sensitivity of the models. This point is also detailed below.*

In the original manuscript of Lombard et al., 2009 (LO09), the errors are given on the individual parameters, and it is not possible (we tried) to reconstruct directly the exact equations used for plotting the 95% confidence intervals, since the parameters are not independent from each other. In order to nonetheless propose an analysis of the effect of the uncertainty on our results, we modified the coefficients of the growth functions used initially as per the figure and table below and analyzed its effect on the depth calibration and the

associated error on the comparison to the MARGO dataset. The chosen range of values is close to the 95% range of LO09 for most species and larger than the 95% confidence interval for the others. The values obtained are thus a maximum range in all values given.

Regarding depth calibration, we find our results to be largely insensitive to the use of these upper-bound and lower-bound values for the growth functions. Specifically, the uncertainty in the maximum growth depth is largest on *N. pachyderma* (range of 475 to 600 meters) and *G. bulloides* (400 to 450 meters). It is somewhat smaller for *G./T.* sacculifer (100 to 125 meters) and *N. incompta* (60 to 65 meters). There is no impact for *G. ruber*. If keeping depths constant and computing the impact of the growth function on the mean difference between simulated and MARGO d18O values shown in Figure (2), the resulting change is lower than 0.1 per mil for each individual species.

This additional analysis shows that our results are very robust.

Action taken: We have added an additional supplementary figure in the manuscript showing the ranges used for the error analysis and added corresponding text to highlight further the impact of using these other, within error curves, on the computation of the d18Ocalcite.

*Technical comments :*

*[l. 18 page 1]: Expand the connection between the stratified nets and the isotopic derived values of Emiliani more in details. In Jones, there is no reference whatsoever to any isotopic analyses. You are making the connexion, but this was not put forward by Jones.*

The line the reviewer is referring to does not say that Jones used isotopic analysis: "Through in situ water column sampling via opening-closing plankton nets, (Jones, 1967) corroborated the depth habitats of Emiliani (1954)." We hence interpret this as a need for clarification in this section of the text.

Action taken: To clarify this point, the statement is altered in the revised version of the manuscript and now reads: "Through in situ water column sampling via opening-closing plankton nets, Jones (1967) through faunal abundance counts corroborated the depth habitats that Emiliani (1954) inferred through isotopic analysis."

*[page 2]: "It is this tempting to make one additional step": I do not understand this statement : It is a fact that oxygen isotopes have been implemented in models, but yet, this is not the topic of the paper as the d18O values of seawater are computed from empirical basin correlations between d18Osw and salinity, not from water isotopes enabled models. I would recommend to move this sentence in the perspectives, as it is misleading here and one quick reader might think that isotope models were used.*

Action taken: we have followed the recommendation of the reviewer and removed this sentence.

*[equation 1, page 3]: I would add here a reference to the original work (Kooijman 2000) which formalized this equation as referred in Lombard et al (2009).*

Action taken: We have added the reference to Kooijman, 2000

*[line 5, page 4]: The authors do use a Tl of 280 for G. bulloides instead of 281.1. This shows that the model is extremely sensitive to a minor change in Tl : could you please elaborate a bit on the reason on this 1.1∘K shift ? Did you perform some sensitivity analyses to reach this temperature ? This is appealing because the overall inferred isotopic equilibrium depth calculated for this species is off the charts (see point below).*

**As explained in l. 5-9, p. 4, comparing FAME's output with the subpolar North Atlantic sediment trap data published in Jonkers et al. (2013), we found that the nominal value of *G. bulloides* lower boundary of the growth tolerance range, TL = 281.1 K, was too high and was responsible for the absence of growth outside of the 3 summer months. In contrast, sediment trap data indicate that, on average over the four years of observations, significant *G. bulloides* fluxes prevailed from the end of June to the middle of November. We tested a few other values for TL and chose the value closest to the nominal value that allowed the extension of the growing season into the fall, in agreement with the data pattern.**

Action taken: we have revised the text in the area of former lines 5-9, page 4 to hopefully arrive at a better formulation. The revised text now reads:

"In the present study, we use the nominal values of equation (1) parameters given in Lombard et al. (2009) with the exception of TL for *G. bulloides*. Indeed, comparing the output of FAME with sediment trap data from the subpolar North Atlantic (Jonkers et al., 2013) showed that the nominal value of TL = 281.1K was likely too high, causing an absence of growth outside of the 3 summer months. In contrast, subpolar North Atlantic sediment trap data indicate that, on average over the four years of observations, significant *G. bulloides* fluxes prevailed from the end of June to the middle of November. We hence chose a value of TL closest to the nominal value of Lombard et al. (2009) that would allow the extension of the growing season into the fall in agreement with the data pattern. Hence a value of TL = 280 K was used for *G. bulloides* within FAME."

*[Figure 1 page 4]: Fig 1 - I would add the original data as in this figure we lose the range of amplitude observed in cultures*

**Adding all the data into Figure 1 results in a rather messy figure as can be seen from the new Appendix figure A1, added nonetheless. Rather than reproducing separated panels as in the figure of Lombard et al. (2009) in the main text, we prefer to show how the different growth curves differ between species.**

Action taken: We have added a figure to the Appendix that takes into account the comment of the reviewer for both the error analysis and the original data of Lombard, 2009, see new figure A1 as below.

[Figure]

*[Figure 1 page 4]: Add reference to Lombard et al., 2009 in the figure caption*

Action taken: We thank reviewer #1 for pointing out this omission. We have corrected it.

*[page 7] Calculation of the best-fitting maximum depth:*
- *What is the rationale for assuming that the Late Holocene equilibrium isotopic value value would be the maximum depth in the model ? Do you imply that the isotopic signature of foraminifera is biased toward the maximum calcification depth ?*

**We do not imply that the isotopic signature of foraminifera is biased toward the maximum calcification depth. In the formulation of FAME, planktonic foraminifers have the possibility to grow anywhere between the sea surface and a so-called "maximum depth" (see equations (7) and (8)). This maximum depth of growth is an additional model parameter for each species. As explained in section 2.3, we have determined the values of these maximum depths of growth parameters, such that the agreement between computed and measured MARGO core top d18O is optimal. In doing so, we define the version of FAME that provides the best fit to core top isotopic data under present-day conditions. This approach is consistent with our goal of developing a module enabling model-data comparison with isotopic records from marine sediment cores (see our answer to reviewer #1's first concern).**

Action taken: We have added the following sentence to section 2.3 to clarify this: "The rationale behind this choice is to specifically design FAME to enable model-data comparison with isotopic records from marine sediment cores."...

- *The range of the depths calculated by the model are very deep compared to observed living depth. The most extreme case is G. bulloides: if one uses the last textbook written by R. Schiebel & Hemleben (Modern planktonic foraminifera, 2017) "Ecology: Globigerina bulloides mainly dwells above the thermocline within the upper 60 m of the water column, and is a non-symbiotic species". The ecology of this species is extremely problematic, and likely due to a combination of multiple cryptic species (eg Morard et al), I would tend to think that the cultures did not catch the overall variability in the dataset.*

   **We agree with reviewer #1 that the ecology of G. bulloides is problematic and that this difficulty is likely related to the existence of multiple cryptic species. There is indeed a wide variety of living depths reported in the literature. Note that our results are in good agreement with the observations of Rebotim et al. (2017) who provide very interesting data from vertically resolved plankton hauls in the subtropical eastern North Atlantic showing that living G. bulloides are found down to 300 m, with a median living depth of ~90 m.**

   Action taken: We have added 2 columns to Table 1 in order to document the observed living depths of the different species: column 6 gives the range of observed living depths for each species and column 7 lists the corresponding references. In addition, we have added a third column (column 5) giving the range of depths of maximum growth computed by FAME and shown in Fig. 4.

- *I do not understand how does G. ruber has a living range reaching +∞.*

   **+ ∞ is the mathematical result, which corresponds to a maximum living depth of 0 m in the present case. Re-reading the manuscript, we agree with the reviewer that the notation was confusing.**

   Action taken:We have modified this in Table 1 to simplify.

   *It would be extremely useful to have a figure putting into context the ranges (by comparing with Rebotim et al, for example event though this is a single figure).*

   **We hope that the information added to Table 1 fulfills this role and will suffice to respond to the reviewers' concern.**

*[lines 18-24 - page 7]: I do agree that those two effects (gametogenic calcite and dissolution) can somehow impact the signature of d18Oc in G. sacculifer. Yet, as G. sacculifer is bearing symbionts, it does have to live in the euphotic zone, which is not the case in the model. I suggest that the authors make a more solid case by removing the deep Pacific sites that they supposed*

*to be influenced by the dissolution to check whether the origin of this deep signature is indeed mostly gametogenic*

**To check this suggestion, we have verified that there is no dependence of G. sacculifer d18Oc on depth in MARGO data set, whether only the Pacific or the whole dataset was used (see figure below). Hence, it is unlikely that there is a dissolution signal in the d18Oc from G. sacculifer from MARGO. Also, the updated data added to Table 1 in response to previous comments shows that the calculated maximum depth of growth for G. sacculifer is in agreement with observations. Most notably, Rebotim et al, 2017 reports G. sacculifer being found alive down to 100 or 200 meters.**

[Figure]

Action taken: we have removed the suggestion that the signal could be due to dissolution in the text of the revised version of the manuscript, previously on p. 7 (l. 20-24)

*[lines 28-30 - page 7]: As the error scheme does not include the error linked to the calibration of the FORAMCLIM model. It would be extremely interesting to have an idea of the sensitivity of the FAME model to the max/min range observed in the data set*

**We do not use any calibrated relationship from the FORAMCLIM model, but simply the growth rate functions derived from culture measurements of Lombard et al., 2009. Here, we suppose that this is what the reviewer refers to by "*the calibration of the FORAMCLIM model*".**

Action taken: As described in the response to the reviewer's second concern, we have derived alternative temperature curves from the Lombard et al., 2009 equations that cover a large range of growth rates and are close to the error curves given in Lombard et al. 2009's original contribution, now presented in the new figure A1. The impact of using such curves is very limited on the computed maximum depth range, and hence on the d18Oc computed. We have modified the text in the manuscript to account for the new figure and included a discussion of this error propagation.

*[line 31 - page 7]: I disagree with the statement that G. sacculifer and G. bulloides can be called "deeper dwellers". The output of the model does rank them as deeper dwellers, but out at sea, they do live mostly in surface to subsurface layers of the ocean (see for example Schiebel and Hemleben, 2017)*

**We agree with the reviewer that the term "deeper dwellers" is probably inadequate in this context, even though G. sacculifer is reported to be found alive up to 100-200 meters (Rebotim et al, 2017).**

Action taken: the term "deeper dwellers" has been removed from the revised version of the manuscript. Ranges found in the litterature have been added to the Table 1 to clarify where FAME stands with respect to what is found in the present-day oceans.

*[Table1 - page 8]: The range is definitely too deep for G. bulloides (ibid.)*

**See our answer above.**

*[Figure 3 - page 10]: I do not really understand what the figure does show : a percentage is highly depending of other species percentages – see my main comment #2. What is the rationale for the cutoff at 10% ? I do not see a physical nor biological rationale for this cutoff. I am wondering if the spatial coverage in the Indo Pacific Ocean is good enough to be included in the analysis as most core tops come the Atlantic Ocean.*

**We thank the reviewer for putting forward this point. As mentioned above, the revised version has been modified to improve the comparison between our presence/absence results and the abundances from the MARGO core top database. An example of such updated figure can be found below. The text of the manuscript has been modified accordingly.**

[Figure]

*[Figure 4 - page 11]: Consider changing the color scheme- rainbow does not give the best rendition.*

Action taken: we have followed the advice of the reviewer and modified the color scheme. We hope it provides a better rendition of the results. The updated version looks as follow.

[Figure]

*[page 12]: Add a latitudinal/depth plot, it would be more easy to read.*

We do not quite understand what the reviewer means here. A latitudinal/depth plot of the depth referred to would merely give a white rectangle with a single line in it, giving the depth of maximum growth. Since a latitudinal/depth plot will require some form of averaging over longitudes, it will be equivalent to figure 5 albeit loosing the longitudinal contrasts. Since this does not seem very helpful, we assume that the reviewer had something else in mind?

---

## Author Comment (AC2) · 27 Jun 2018

*Responses to Anonymous Referee #2*

**NOTA: The initial reviewer comments are in italic, our answers are in bold, action taken in the revised version of the manuscript are underlined.**

*The manuscript by Roche et al. summarises a noble and interesting attempt to improve our understanding of foram-based oxygen isotope data. The authors present a module ('FAME' – Foraminifers As Modeled Entities) they developed in order to predict changes in the oxygen isotope composition of the tests of different foraminifera species in response to changing climatic conditions. The model is forced by hydrographic data alone and incorporates a limited number of species-specific parameters, based on culture experiments, for each of five foraminifera species to describe their growth and habitat. Essentially, the model attempts to account for the effect of foraminifera depth habitat on their oxygen isotope composition, and to predict their oxygen isotope composition accordingly, as well as their presence/absence. To test their model they apply its methodology to reference datasets, namely the MARGO Late Holocene dataset. It is an interesting and concise presentation of their work and well-structured. I believe it will greatly contribute to research within the foraminifera and palaeoclimate community.*

**We are thankful for these positive and nice words on our work.**

*When such models are developed it's important to have some measure of their sensitivity. For that reason, I believe that error propagation in the model should be addressed given that several of the input parameters have errors associated with them.*

**We thank the reviewer for this remark. In response to this concern and to a similar concern expressed by the other reviewer, we have now included a discussion of the propagation of the errors in the Lombard growth equations into the maximum depth calculations and further in the distribution error of the previous figure 2, giving computed error arising from the propagation of the initial input parameter errors.**

Action taken: In response to the reviewer and the other reviewer who had a similar request, we have tested a large range for the Lombard et al., 2009 growth rate curves and propagated the error in our results. The range tested are now presented in the new figure A1. Additions were made in the text where appropriate to describe these tests. The main message is that our results are very robust to such changes with an impact on the mean difference between computed and MARGO d18O below 0.1 per mil for all species.

*Secondly, and this may sound pedantic but the authors may consider changing Globigineroides sacculifer to Trilobatus sacculifer as per its genus reassignment by Spezzaferri et al. (2015). I will leave this to the authors' discretion as there are arguments for retaining G. sacculifer given that this is still the most commonly used name for this species. However, over time this will obviously*

*change and the authors may want to introduce the new (and more taxonomically up-to-date) name.*

**We thank the reviewer for this comment that we have taken into account as suggested.**

Action taken: The revised version makes reference to *Globigerinoides sacculifer* as the forma name (including *Globigerinoides trilobus*) in the first instance and uses *Trilobatus sacculifer* throughout.

*In terms of convention, there are several instances where the author refers to oxygen isotopes incorrectly. For example, p1, line 16, the authors describe the 'oxygen-18 value', or in line 21, 'calcite oxygen-18', or elsewhere as 'species' oxygen-18' (e.g. p3, line 1). This is very pernickety but there are quite strict guidelines for isotopic notation. I suggest the authors double check their usage and perhaps refer to ratios rather than oxygen-18 content/signal as it's more in line with the literature.*

Action taken : we have checked thoroughly the revised manuscript for mis-use of oxygen isotopes and have corrected them following the suggestion of the reviewer.

*On page 7, line 16-17, you describe how you used a 0.1 per mil 'encrustation term'. Could you possibly elaborate as to where that value came from? It would make it easier for the reader as it seems a little arbitrary at present.*

**We have chosen a 0.1 ‰ value for the encrustation term in order to simulate maximum depths in agreement with the literature. The simulated depth of maximum growth shown in Fig. 4 and now summarized in the revised Table 1 do indeed match very well the available observations. For instance, Fig. 4e shows a deepening of *N. pachyderma* depth of maximum growt from 0-30 m in the Greenland Sea to 100-350 m in the Norwegian Sea, in agreement with the apparent calcification depths reconstructed by Simstich et al. (2003).**

Action taken : we have added a few lines to summarize this to the text accompanying Table 1.

*Also, the authors should mention wherever necessary that species with symbionts e.g. G. sacculifer (T. sacculifer) cannot live at depths greater than the photic zone, as is hinted at on page 7, line 20.*

**We agree with the reviewer that it is important to explicitly mention that T. sacculifer bears symbionts, like G. ruber. We would like nonetheless to emphasize a few aspects. Experimental determination by Spero suggests that the removal of symbionts drastically reduces the life-span of the host, therefore the reviewer is correct that symbiont bearing foraminifera should ideally inhabit the photic zone. However, the irridiance required for these**

symbionts is not known and hence we cannot infer maximum depth from that perspective. And finally the host is known, prior to gamete release, to be symbiont barren therefore there is a portion of growth that can be below the photic zone.

Action taken : we have modified the text p. 7 to explicitly mention that T. sacculifer bears symbionts.

*Some more specific comments:*

Action taken : specific comments have been corrected following the suggestions of the reviewer but for one instance.

*Page 1, line 17.  Perhaps use 'reflected' rather than 'favoured'.*

*Line 20.  Use 'throughout the year' rather than 'along the year' as this makes more grammatical sense.*
OK
*Page 2, line 7.  I would consider adding a few more references here as several other studies have been done looking at carbonate ion and symbiotic effects. Pearson et al. (2012) gives a good summary of work up to that point.*
OK
*Line 26.  Change 'being' to 'to be'*
OK
*Page 3, line 19.  Italicise N. pachyderma.*
OK
*Page 5,line 21. Change 'weighs' to 'weight'?*
No, weighs is what we mean here.

*Page 8, line 2. Use a different word to 'ascertain' as this doesn't make sense in the context.*
We replaced it by « check »